# The Impact of Rail Transit on Accessibility and Spatial Equity of Public Transit: A Case Study of Guangzhou, China

**DOI:** 10.3390/ijerph191811428

**Published:** 2022-09-11

**Authors:** Huiling Chen, Wenyue Yang, Tao Li

**Affiliations:** 1School of Public Administration and Human Geography, Hunan University of Technology and Business, Changsha 410205, China; 2College of Forestry and Landscape Architecture, South China Agricultural University, Guangzhou 510642, China; 3Northwest Land and Resources Research Center, Shaanxi Normal University, Xi’an 710062, China

**Keywords:** rail transit, public transit, travel time, transit fare, accessibility, equity

## Abstract

The urban rail transit network provides the possibility for people to shift from car to public transit for travel. This paper clarified the relationships among public transit, accessibility, and equity and studied the impact of rail transit on public transit accessibility that incorporates the measure of travel time and transit fare and the impacts’ spatial equity. The results show that rail transit contributes to the similar distribution between high rate of changes of time-based accessibility communities and fare-based accessibility communities, which are located nearby the rail transit lines. The degree of inequity in travel time is higher than the degree in transit fare in two scenarios. Due to the well-connected bus transit in the city center, absolute changes in travel time are slight, while relative changes are high. The rail transit has promoted the improvement of public transit equity in some areas. The difference between the time-based accessibility of Conghua District, northern and southern Baiyun District, Huadu District, Nansha District and southern Panyu District, and other communities is getting smaller, which is conducive to the improvement of spatial equity. The results provide theoretical support for the development of an integrated multimodal public transit system.

## 1. Introduction

Conventional bus transit with limited traffic capacity cannot meet the public transit needs of big cities due to rapid expansion of the city, widening activity spaces, and lengthening travel distance [1,2]. The building of rail transit infrastructure has already provided an effective supplement to the public transit system. Urban rail transit is gradually dominating the public transit system due to characteristics of large capacity, speediness, reduced air pollution, and land use and is regarded as an important way to solve urban traffic congestion and achieve energy-saving and emission reduction targets [3]. However, the expensive construction cost of rail transit leads to its limited coverage rate in the transit network, which fails to meet the social needs of those dependent on public transit. From the economic view of travel cost, there is a gap between rail transit fare and bus fare. Individual consumption ability impacts the transport mode choice of riders, which affects the accessibility level. A household travel survey in Canada in 2001 showed that approximately 78% of households that earn less than $40,000 commute by public transit, among which only 13% of such income groups are rail commuters; the majority of people use a cheaper method of bus transit [4].

Urban rail transit construction in China is in progress. By 2017, 31 cities opened an underground railway, among which Shanghai has the greatest mileage. Studying the impacts of rail transit networks on the improvement and disparities of transit accessibility could help plan new rail transit lines and stations and help them coordinate better with the existing transit network, which attempts to increase the transformation possibility of the travel mode choice of the population from car to public transit [5]. This kind of research has an important implication for the improvement of the urban transport service level, the satisfaction of the diverse and growing public transit demands, and the realization of sustainable urban transport development. Rail transit significantly reduces the travel time between communities, generating both efficiency and spatial equity effects. It also leads to the obvious “time–space convergence” effect, enlarges the influence of the city center, creates transport advantage for more areas, reconstructs the urban structure, and influences the spatial distribution of economic activities [6,7]. However, it aggravates disparities of accessibility among regions and brings about the concern of transport equity [8,9,10,11].

Accessibility indicators are increasingly being used as supporting tools for transport infrastructure planning. Our study contributes to the existing literature by evaluating the influence of rail transit on transit accessibility and by answering the two following questions: (1) What are the spatial characteristics of the travel expense and travel time of communities by bus transit or integrated bus-rail transit and are there significant spatial differences between the two travel modes? (2) How are disparities for transit accessibility based on the rail transit network distributed across geographical areas and combined under an equity objective? The remainder of the paper is organized as follows: Section 2 presents the literature review on transit accessibility equity. The study area, data, and methodology are illustrated in Section 3. Section 4 evaluates the spatial influence of the rail transit in Guangzhou. The final section demonstrates conclusions and discussion about future research directions.

## 2. Literature Review

Equity, regarded as a policy target in public transit supply, initially appeared in the 1990s. One way to achieve goals about equity in the city is to provide transit services to the people who need them most frequently. Equity is a concept that involves multiple dimensions and multi-faceted aspects, and geographers mainly study equity in the distribution of public service facilities. Litman (2007) argued that public transport equity analysis is supposed to concentrate on two dimensions: horizontal equity and vertical equity [12]. Horizontal equity emphasizes the distribution of transport resources among various individuals. The studies that are related to the horizontal equity of public transit explore the minimum equity standard of “uniform” people without considering the socio–spatial differentiation. Meanwhile, vertical equity refers to the distribution of public transit resources among individuals differing in capabilities and needs. Thus, it emphasizes the needs of various social groups to alleviate social inequity. The allocation of public transit resources is a process of meeting the daily travel needs of people. The study on the equity of public transit has gone through three stages: geographical equality, spatial equity, and social equity. The studies on the spatial equity of public transit focus on the issue of equivalence, which is the minimum standard of “homogeneous persons”. The studies on social equity began to pay attention to the in justice between different social groups.

The equity assessment of public transit supply is closely related to the measurement of accessibility [13,14]. Accessibility conventionally is defined as potential opportunities for interaction [15], which is an indicator commonly adopted to measure the geographical effects of the transport network and the relevance between transport and economic growth and to evaluate the impact of urban transport on travel and the land-use pattern [16,17,18]. Gleason (1975) analyzed a dataset covering bus stop locations by virtue of integer programming models, marking the beginning of studies on transit accessibility [19]. Existing studies on transit accessibility mainly focus on the measurement of accessibility and its application in different cities, among which the differences in the function adopted in accessibility application make the measurement differ significantly. There are three types of indicators for transit accessibility. The first type of indicator refers to physical accessibility based on a European geometrical distance, in which the common indicator is the walking distance from the starting site to the transit stop [20]. Such indicators are simple and easily calculated. However, they disregard the service provided by each transit station, desired destinations, and travel time to these destinations, so the accessibility reflected by such indicators is incomplete. Compared with the aforementioned indicator, the second type of indicator evaluates the service frequency of transit stations, which means that the importance of the station in the entire transit network has been considered [13]. The third type of indicator additionally accounts for the travel cost associated with destinations [21], as well as the temporal variability of public transit [22,23]. General Transit Feed Specification files, which construct a multimodal public transit network, are an important technical improvement. A Google Transit Feed Specification (GTFS) dataset has been recently applied to build a completely routable multimodal transit network, which enables the estimation of public transit travel times at different times of the day [24,25]. The fourth type of accessibility indicator refers to the log sum of various opportunities within the specific cost calculated from a regional travel demand model under public transit travel situations [26]. Most of these indicators are used for studies on the spatial equity impact of public transit. The social equity of public transit studies mainly focus on the limited accessibility issues that are affected by different individuals’ space–time constraints [27]. Existing studies often firstly used specific criteria to distinguish transport-disadvantaged groups and then discussed the social equity of their travel behavior [28]. On the one hand, some studies explore the various impacts of public transit on the travel behaviour of residents in different regions, and it is unfair that some have fewer travel opportunities and transport resources than others [29]. On the other hand, some researches analyzed the relationship between the travel behavior of transport and the disadvantaged groups’ socio–economic attributes under the influence of public transit accessibility [30].

In the studies reviewed above, several authors incorporated public transit travel expenses into the measurement of transit accessibility due to complex fare structures. Ford (2015) and Currie (2004) measured transit accessibility based on formal costs, which include travel time and travel expenses [31,32]. El-Geneidy et al. (2016) assessed the differences in accessibility resulting from transit fares by considering monthly or single fares [33]. The approaches they used to calculate the fares were not applicable to other rail transitpolitan areas because of the data. Different from the diverse ticket pricing strategies in many countries, the prices of public transit tickets in China are typically determined by the government. Although some special riders enjoy privileges (such as elderly individuals and students), the rest are charged the same when taking bus transit, and prices are based on travel distances for rail transit. To our knowledge, most authors have analyzed the positive effects of rail transit on increasing land values and accessibility in travel time [34,35], and literature on the multimodal public transit system mostly focuses on the joint optimization of rail transit lines and bus transit lines as well as the distribution of transfer stations [36,37,38], but the assessment of the spatial equity impacts caused by rail transit is limited. Therefore, Guangzhou is a typical empirical case for the study of the effects of the rail transit on public transit accessibility.

In this paper, travel time and transit fares are incorporated into the measurement of transit accessibility. We study disparities of the accessibility changes among communities under the influence of the rail transit and evaluate its spatial equity effects in an attempt to offer theoretical support to the development of an integrated multimodal urban public transit system.

## 3. Methodology

### 3.1. Study Area

This paper covers the study of Guangzhou city’s 11 municipal districts, including an area greater than 7434.4 km^2^ as well as more than 2000 communities (Figure 1). As one of the largest cities in China, Guangzhou city is located in the Pearl River Delta economic zone, known as a member of three significant and domestic economic circles. It is famous across the world for its leading role in the reform and opening-up to the rest of the world. Undoubtedly, Guangzhou city reflects the development of important cities in China to some extent. Therefore, we employ community-level statistics about the population gained from the sixth national population census results of Guangzhou city in 2011. The destinations of all trips are 11 major commercial complexes in Guangzhou. The type of destination is not that important at this point, as the basic aim of this paper is to analyze the impacts of rail transit on the accessibility by public transit—therefore, spatial distribution is the major criterion for their selection rather than the category of facility. Major commercial complexes are located evenly in the study region and thus provide a relatively typical sample for this study.

### 3.2. Data and Accessibility Calculation

The navigation function that the Baidu map provides automatically can implement the travel schemes between any two geographic points in the city based on the travel mode choice. We made iterate inquiries with the WebAPI based on the Baidu map LBS open platform in Pythonscript. Real-time transport information for the Baidu map is estimated every two minutes based on data from multiple sources, considering not only the speed of different levels of roads but also real-time traffic congestion, which can render an estimation with a high accuracy of travel distances and travel time [39,40,41]. The road network of the Baidu map is complete and updated constantly, in which travel distances and travel times are calculated with the total for all segments along the route [36]. To reflect the reliability of travel data based on the Baidu map, we invited 10 residents who live in different communities to record their daily travel time and travel distances on public transit (including the use of rail transit or bus and transfers involving rail transit and bus), while the same period of time and same distances were calculated with the Baidu map. Fifty tests were performed. The results show that the difference between the actual travel time/distances and those calculated based on the Baidu map ranged between −10% and 10%, so the travel time and travel distances based on the Baidu map can be considered reliable.

We took the centers of communities as the origins and 11 major commercial complexes as the destinations. We found the best paths between the starting points and the ending points using two different travel modes—the mode of the bus transit that is based on the bus transit system and the mode of the shortest travel time that is based on the integrated public transit system, here named the bus-only scenario or bus and rail transit scenario, respectively. The multiday and multiperiod methods were used for data collection to obtain the daily travel time and travel distance by public transit. The acquisition time included five working days and one weekend, and the average was calculated to represent the daily travel time and travel distance. The data of travel distances, travel times, and travel routes information between all communities and major complexes were obtained between 12 November and 19 November 2020. The dataset included 409,794 routes, 409,794 records of travel time, and 409,794 records of travel distance for each scenario.

As for the bus and rail transit scenario, the travel scheme with the shortest time would be selected if the rider could reach the destination by bus or rail transit directly. The scheme with the shortest total time would be selected if the rider could reach the destinations by rail transit involving transfers or they could reach the destination by bus transit directly. The scheme with the shortest total time would be chosen if the rider cannot reach the destination by either direct bus or rail transit. For the bus-only scenario, the travel scheme with the shortest time would be selected. Most regular bus transit fare is 2 yuan, and a few bus express lines connecting the main urban area with the marginal urban area (Zengcheng District, Conghua District, Nansha District) have higher prices, ranging from 10 yuan to 20 yuan. The ticket for rail transit is priced based on travel distances by rail transit. We first compared the travel time, transit fare, and travel speed among the CBD, major commercial complexes, and all communities under different scenarios. The impacts of rail transit on the equity of the spatial distribution of travel times improvement and increased transit fare cost were then discussed. The calculation of travel costs in different scenarios is shown in Figure 2. Residents who travel by public transit generally included three stages: walking from the starting point O to the public transit station stage, taking public transit to the station near the destination stage, and walking to the terminal D stage. The travel time and travel fare of these three stages were summed in this paper, and the time spent during transfer in the second travel stage was also considered.

This paper analyzed the accessibility changes based on the average travel time and transit fare from communities to major commercial complexes between the bus-only scenario and the bus and rail transit scenario. The expression is as follows:$$T_{i}=\sum_{j=1}^{n}t_{ij}/nP_{i}=\sum_{j=1}^{n}p_{ij}/n$$
where $T_{i}$ and $P_{i}$ are the accessibility of community i, $t_{ij}$ is the travel time to the destination of the commercial complex j, and $p_{ij}$ is the transit fare to destination j. $t_{ij}$ and $p_{ij}$ adopt the minimal travel time or transit fare. The reduced value of $T_{i}$ in the bus and rail transit scenario denotes the travel time saved in community i, and the increased value of $P_{i}$ denotes the transit fare raised in the community. The community with the lowest average travel time or transit fare is considered to have the highest time-based accessibility or fare-based accessibility level among all communities.

### 3.3. Equity Analysis

The equity analysis in this paper is based on the differences in the spatial distribution of accessibility improvement between these two scenarios. Equity effects are usually measured by using a range of indicators of the spatial distribution of accessibility indicators. The selection for these indicators is based on the changes in accessibility between scenarios. An ideal equity indicator does not exist, and many scholars advise calculating a set of indicators to analyze their results as a complement. We utilized three steps for evaluation of the equity effects based on the findings. In the first analysis, the coefficient of variation is the ratio of the standard deviation to the average, which represents the relative change of the geographical data. The formula for *CV* is:$$CV=\frac{1}{x}\times \sqrt{\frac{\sum_{i=1}^{n}{\left(x_{i}-x\right)}^{2}}{n-1}}$$

In similar research, this indicator has been used frequently for the purpose of evaluating the equity effects. The increased *CV* value states the reduction in equity and negative equity effect, whereas a reduction in the *CV* value indicates the positive equity effect and more balanced spatial distribution of accessibility.

Then, the normalized value of relative and absolute accessibility improvement were calculated in each community. These two values are complementary because a community is able to obtain an absolute improvement but a relative low improvement if its initial accessibility value has a low level. The value of the absolute improvement and relative change ratios of travel time of communities after the operation were normalized (z-score) in this paper so that results could be comparable. A z-score of 0 is equivalent to the average accessibility benefit of the entire city. Accessibility benefits could be completely equivalent throughout the city if all communities have scores of 0. Positive values indicate a community has a larger than the city average in accessibility benefit, and negative values show it as lower than the average accessibility benefit. We illustrated how the values are distributed across each district to better assess the equity of accessibility changes.

Finally, we identified the role of different communities in the spatial equity of the rail transit by comparing two indicators referring to time-based accessibility and the rate of change between the two scenarios, which is beneficial for the subsequently targeted improvement of the regional transportation infrastructure and the promotion of spatial equity of transport accessibility. All communities were divided into two categories: ① The differences between the two scenarios tended to be smaller, which were conducive to spatial equity; this included two situations: low time-based accessibility (travel time higher than the average)—high rate of change (rate of change higher than the average), high time-based accessibility (travel time lower than the average)—low rate of change (rate of change lower than the average); ② The differences between the two scenarios tended to be larger, which was not good for spatial equity; this included two situations: low time-based accessibility (travel time higher than the average)—low rate of change (rate of change lower than the average), high time-based accessibility (travel time lower than the average) and high rate of change (rate of change higher than the average).

## 4. Results and Analysis

### 4.1. The Impacts of Rail Transit on Accessibility

#### 4.1.1. CBD Accessibility Differences

From the selected origin (the commercial complex in the CBD) to all communities, the differences in the reachable area and population within periods, and the average travel speed and transit fare between the bus-only scenario and the bus and rail transit scenario are illustrated in Table 1 and Table 2. In the bus-only scenario, 107 communities can be reached within a 30-min drive, and 637 communities can be reached within a 1-h drive. Within the same time, 476 and 1245 communities are reached in the rail transit scenario. The spatial coverage area and inhabited population within a 10- to 30-min isochronous circle from the CBD in the bus and rail transit scenario are more than double compared to that in the bus-only scenario (Figure 3a,b).

In the bus-only scenario, the isochronous circle of the CBD is distributed in a concentric mode. In the rail transit scenario, the isochronous circle of the CBD is extended and spread along the rail transit lines in a fingerlike style. Figure 3c shows the spatial distribution of speed ratios between the two scenarios in the communities. The maximum travel speed ratio is 3.75 when evaluating trips from all communities to the CBD. Communities with small differences in travel speed are found in the southern Nansha District, eastern Huadu District, most of the Zengcheng District, and part of the central city. Communities with a high variation of travel speed are near the rail transit line. There is a mixture of high ratio communities and communities with the ratio close to 1 in the city center. The fastest travel speed from communities to the CBD by rail transit is 3.75 times the speed of bus transit. The speed of travel in different scenarios significantly varies for communities along rail transit networks.

#### 4.1.2. Spatial Distribution of Time-Based Accessibility

We calculated the travel times from all communities to 11 major commercial complexes. The operation of rail transit lines leads to a decrease in mean time from 31.58 h to 19.68 h, and the total travel time decreases from 83,945.71 h to 52,314.46 h. The improvement of the total travel time is 37.68%, showing that the rail transit greatly reduces the time between communities and commercial complexes and promotes the communication of social economy and culture. As Figure 4a reveals, the spatial pattern of time-based accessibility shows the inner city is the core center with a low-value spatial distribution, and the value increases from the core to the fringe area. The spatial pattern of time-based accessibility shows the north–south extension in a “concentric” shape. The community with the longest average travel time is 8.65 times that with the shortest average travel time.

The spatial pattern of time-based accessibility is slightly reversed with the rail transit network (Figure 4b). The overall level of time-based accessibility in the city center is always the highest, indicating that the average time of reaching all major commercial complexes is the lowest, while the fringe areas need more time. The community with the longest travel time (519.42 min) is 11.3 times the community with the shrotest travel time (45.95 min) due to the improved link brought about by the rail transit lines. Figure 4c shows the percentage of changes in the time-based accessibility level between the two scenarios. This clearly highlights the transformation that would occur in communities of the Conghua District, as their time-based accessibility values in the bus-only scenario are low. The communities near the rail transit lines see high improvements, which offer virtually direct access to the commercial complexes. The lowest percentages of improvement occur in the communities of the northern Zengcheng District, which obtained an improvement value of less than 12.67%, demonstrating that they are less affected by the rail transit lines.

#### 4.1.3. Spatial Distribution of Fare-Based Accessibility

Fare cost-based accessibility was calculated and compared between the two scenarios (Figure 5a,b). In the bus-only scenario, the average transit fare for Tianhebei residential block in Tianhe District is the lowest (3.27 RMB), while communities with high transit fares are in the western Huadu District and the southern Nansha District. The average fare between the Zhoudong village of the Conghua district and major commercial complexes is the highest (12 RMB). The highest average fare is approximately three times the lowest fare. In the bus and rail transit scenario, the average transit fare between the Lvhe residential block in the Tianhe District and major commercial complexes is the lowest (6.09 RMB), while the transit fares of communities near the rail transit lines are high. Figure 5c shows the rate of change of the transit fare bbetween the two scenarios. The rail transit significantly affects the spatial distribution of transit fare-based accessibility. The spatial pattern of the changes in the time-based accessibility and transit fare is similar for the two scenarios. Communities with a high rate of change are located near the rail transit lines.

Figure 6 shows the proportion of the population in the areas within certain travel times and fare costs. Approximately 80% of the population can reach the nearest commercial complex within 43 min by bus only with a travel cost lower than 2 RMB and can reach the nearest commercial complex within 30 min with travel costs lower than 4 RMB through combined bus and rail transit.

### 4.2. The Impacts of Rail Transit on Spatial Equity

Equity is one of the social impacts brought about by the construction of transport infrastructures. The calculated *CV* values of time-based accessibility and fare-based accessibility in two two scenarios are used as major indicators for the evaluation of the degree of spatial disparity variability changes. The results are shown in Table 3. The increased (13.95%) *CV* value of the time-based accessibility from scenario 1 to scenario 2 implies that the whole city has departed from the relief of disparity from the travel-time reduction after the development of rail transit. The inner city, middle city, and outer city areas witnessed a significant increase in the spatial disparity of travel time after the construction of the rail transit. The rail transit provides citywide intensification of the spatial inequality of transit fare, as reflected by the *CV* values from scenario 1 (0.27) to scenario 2 (0.28). Clearly, in the bus and rail transit scenario, the equity benefits of travel-fare expenses were received by the inner city and middle city, with the disparity in the transit fare within the outer city intensified.

The *CV* of the time-based accessibility of communities in the bus and rail transit scenario is higher than that in the bus-only scenario, but its average and standard deviation are lower, indicating that the rail transit has greatly reduced the average travel time of communities, and the gaps of services acquired from the main commercial complexes among communities are widening. The standard deviation, the average, and the *CV* of the fare-based accessibility of communities in the bus and rail transit scenario are higher than those in the bus-only scenario, which means that the operation of rail transit increased the disparity of public transit service prices among communities, improving fairness in distribution. By comparing the *CV* values of time-based accessibility and fare-based accessibility, we found that the *CV* value of time-based accessibility is higher than that of the fare-based accessibility in the two scenarios, indicating that disparities in travel time among communities are greater than those in transit fare and the inequity degrees of travel times are higher than those of the transit fares.

Figure 7a,b shows the spatial distribution of Z-scores of the relative improvement and absolute improvement of the travel time in urban communities. The green areas indicate that their accessibility gains are above average and raise the overall equity, whereas red areas illustrate that their accessibility improvements are below average and reduce the overall equity. The communities with percentages of improvement above average occur in Panyu District and Conghua District. Comparing the two accessibility maps, we find that communities in the center of Panyu District, Baiyun District, and Conghua District have Z-scores greater than 0.5 for absolute and relative improvements. Most communities in Liwan District, Yuexiu District, Haizhu District, and Tianhe District have above-average relative improvements in time-based accessibility, and their absolute improvements are below average. This circumstance occurs because these communities have high time-based accessibility levels in the bus-only scenario due to good bus transit connections. Communities in Zengcheng District and western Baiyun District are disadvantaged in the bus-only scenario; both their absolute improvements and improvement rates are at significantly low levels as well. Therefore, the accessibility gap between these communities and the rest is even greater, and the inequity related to rail transit is increased.

Table 4 and Table 5 illustrate how accessibility benefits are distributed across the population of each district, and z-values of the accessibility benefits are used to show the percentage f population within each district that falls into the range of half standard deviation increments. Table 5 shows the distribution of the normalized values of absolute improvements across the population of each district and reveals that the majority of the population in districts of the inner city is lower than the mean of the entire city, whereas the districts of the outer suburbs suggest the opposite. To explore the distribution of relative improvements in relation to the population of each district in Table 5, the analysis results can be compared with those in Table 4, which shows the results of the absolute improvements. Overall, the largest upward shifts toward the mean are experienced by districts in the inner city, in which a large percentage of the total population shifts from below the mean of absolute improvements of the city to above the mean of relative improvements.

Communities in Conghua District and Panyu District have a disadvantaged position in the bus-only scenario. Due to an initially poor position, rail transit lines bring high absolute improvements but low relative improvements to most residents in these two districts. The majority of the population in Conghua District has higher z-value scores for absolute improvements and relative improvements than the average value in the bus and rail transit scenario. The position of the population in Baiyun District relative to the mean is worse, with low absolute improvements and low relative improvements. Notably, 25.79% of Conghua District’s population has relative improvement z-values greater than 1.5.

According to Figure 8, the increasingly larger differences are mainly located in Zengcheng District, southern Nansha District, and Huadu District, which are on the edge of Guangzhou. The time-based accessibility of these communities is lower than the average in the bus-only scenario, and the rate of change is also lower than the average in the bus and rail transit scenario, which widens the gap between these communities and other areas with superior time-based accessibility. On the contrary, central urban areas have high time-based accessibility and a high rate of change, so rail transit also widens the gap between communities in central urban areas and other communities. Increasingly smaller differences are crucially located in Conghua District, northern and southern Baiyun District, Huadu District, Nansha District, and southern Panyu District, among which Conghua District, Northern Baiyun District, Huadu District, and southern Panyu District have low time-based accessibility and a high rate of change. Correspondingly, western and southern Baiyun District and other places have high time-based accessibility and low rate of change. The rail transit reduces the differences in the time-based accessibility between communities of this type and other communities. The spatial equity of time-based accessibility is widely observed in these communities.

## 5. Conclusions

As investments in citywide rail transit projects increase, often accompanied by lofty goals for invigorating economic development, this paper serves to highlight the importance of designing a relatively equitable public transit system throughout the urban area. With the use of a travel data matrix obtained from the Baidu API, this paper examined the influence of rail transit on the public transit accessibility by using travel time and transit fare for all communities of Guangzhou. On this basis, we analyzed the spatial equity impacts of the rail transit and the potential for improving accessibility for the population.

The main conclusions are as follows. (1) In the bus-only scenario, the time circles of the CBD are concentric. Eighty percent of the population can reach the nearest commercial complexes within 68 min, with a fare cost lower than 2 RMB. The travel time and transit fares of communities and their needs are closely correlated. In the bus and rail transit scenario, the time circles of the CBD extend along rail transit lines similar to “fingers”. Eighty percent of the population can reach the nearest commercial complexes within 56 min, a with fare cost lower than 4 RMB. Travel time and transit fares in the community have less of an association. The rail transit significantly lowers the correlation between the transit fare and travel time. (2) The rail transit increases the internal disparities of time-based accessibility that generally increases from the inner city via the middle city to the outer city and widens the gaps in locational advantage and the level of accessibility and public transit service prices among communities and exaggerates unfairness. The degree of inequity in travel time is higher than the degree associated with the transit fare in the two scenarios. (3) The Zengcheng District and western Baiyun District are in the disadvantaged position in the bus-only scenario, where absolute improvements and relative improvements in travel time in the bus and rail transit scenario are lower than the average. Therefore, rail transit widens the gap between communities in these areas and the rest, leading to an increase in inequity. (4) The rail transit affects most residents in Conghua District and Panyu District, whose absolute improvements are high and relative improvements are low. Most of the population in the inner city is below the mean of absolute improvements and above the mean of relative improvement. The position of the population in the Baiyun District relative to the mean is worse, with low absolute improvements and low relative improvements. The rail transit reduces the time-based accessibility differences between communities in Conghua District, the peripheral area of Baiyun District, the center of Huadu District and southern Panyu District, and other communities, where the spatial equity of time-based accessibility is widespread in these communities.

Researching the impact of rail transit on transit accessibility is an efficient approach to assess the spatial imbalance of the given effect. However, accessibility for each individual and the public transit needs of each community are different. Our limitation of this accessibility approach results from its aggregate nature, which reflects general differences in the community level and does not combine the residents’ travel demand. In the future, the coordination of the rail transit network and bus transit network could be analyzed further based on the spatial distribution of residents’ travel demand and urban economic activities.

This study helps us to understand the influence of the distribution of rail transit on the travel behavior of public transit and more clearly outlines the structure of the public transit system of Guangzhou. The difference in the travel structure hides transport unfairness. The difference in accessibility in different travel modes, to a certain extent, determines the difference in transport equity.

The rail transit supply in Guangzhou is concentrated in the central urban area, which will help solve the problem of urban transport congestion and CBD accessibility, but the distribution of public transit has a low density and there is only one regular bus line in many communities in peripheral areas. In the future, the government could improve the existing public transit services in communities with low-level accessibility by providing regular bus lines and small buses connecting to rail transit stations. Future policies could combine the merits of the reliability and speed of rail transit and the wide coverage of bus transit for the gradual optimization of the public transit network by region. The planning of new rail transit stations and lines should emphasize coordination with the bus transit system and could draw attention to the connection with the public bicycle network, so that the influence range of the rail transit network can be increased from 500 m on foot to 2000 m by bicycle, which encourages more people to choose rail transit as their daily travel mode choice. This paper only analyzed the impact of rail transit on conventional bus travel based on the travel cost. In the future, the coordination of the rail transit network and bus transit network could be analyzed further based on urban expansion, the spatial distribution of residents’ travel demand, and urban economic activities.

## Figures and Tables

**Figure 1 ijerph-19-11428-f001:**
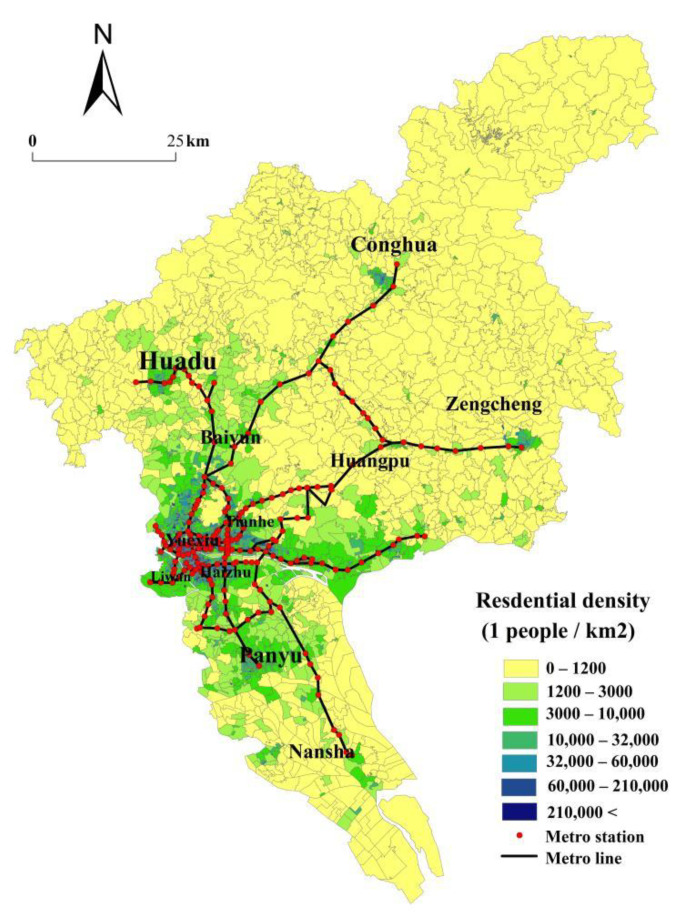
The spatial distribution of residential density and major commercial complexes in Guangzhou.

**Figure 2 ijerph-19-11428-f002:**
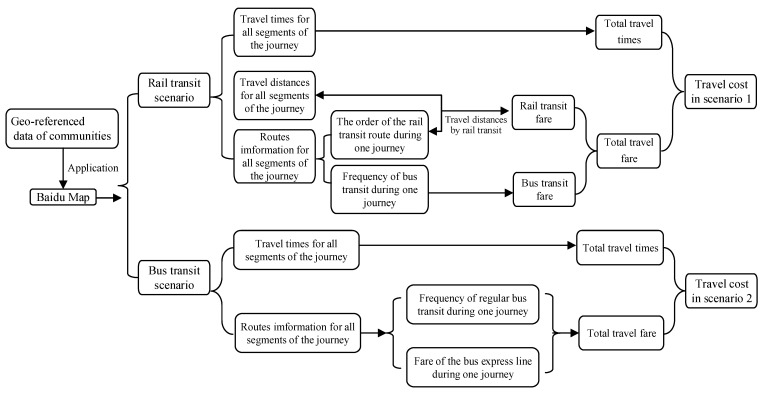
The flowchart of obtaining the OD matrix of travel costs in two scenarios based on the Baidu map.

**Figure 3 ijerph-19-11428-f003:**
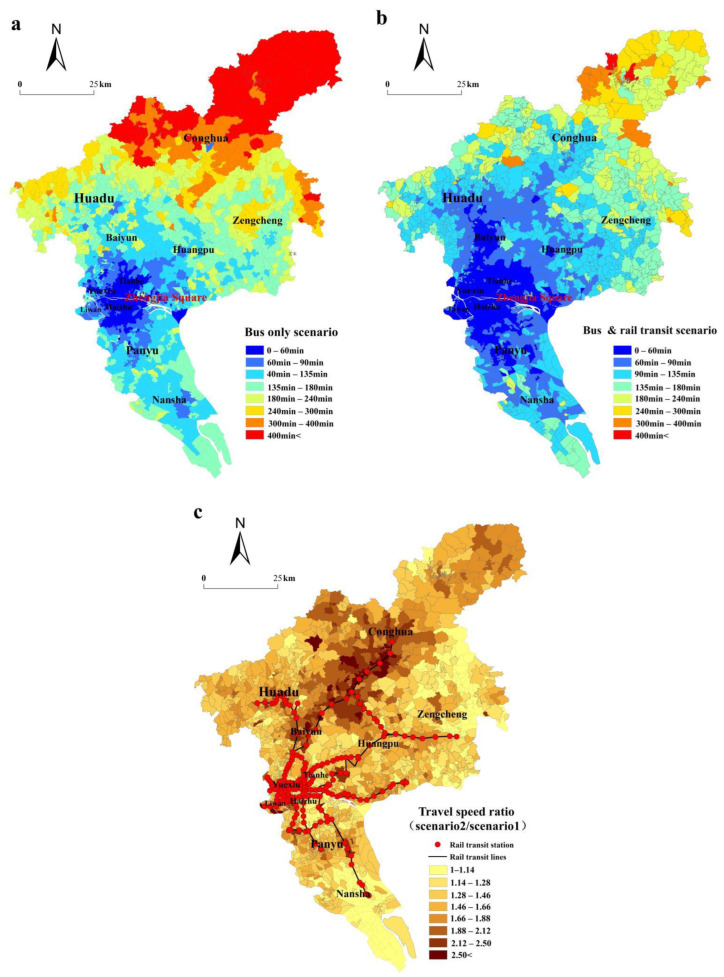
The spatial distribution of time-based accessibility of the CBD in the bus-only scenario (**a**), in bus and rail transit scenario (**b**), and the average travel speed ratio between the two scenarios (**c**).

**Figure 4 ijerph-19-11428-f004:**
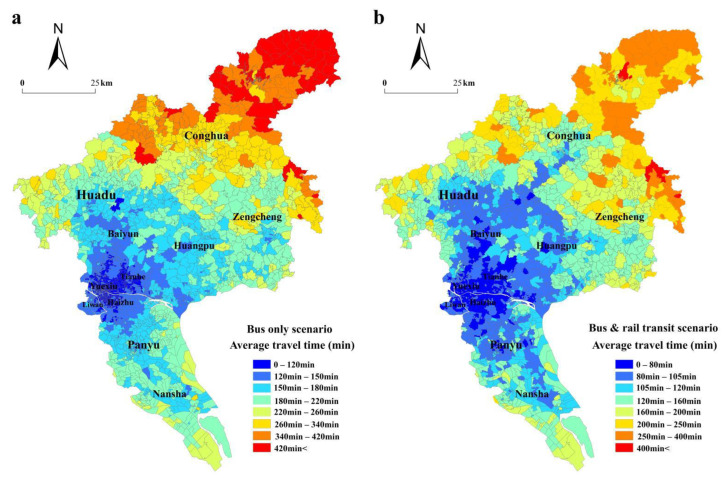
The spatial distribution of the time-based accessibility of communities in bus-only scenario (**a**), in bus and rail transit scenario (**b**), and accessibility changes between the two scenarios (**c**).

**Figure 5 ijerph-19-11428-f005:**
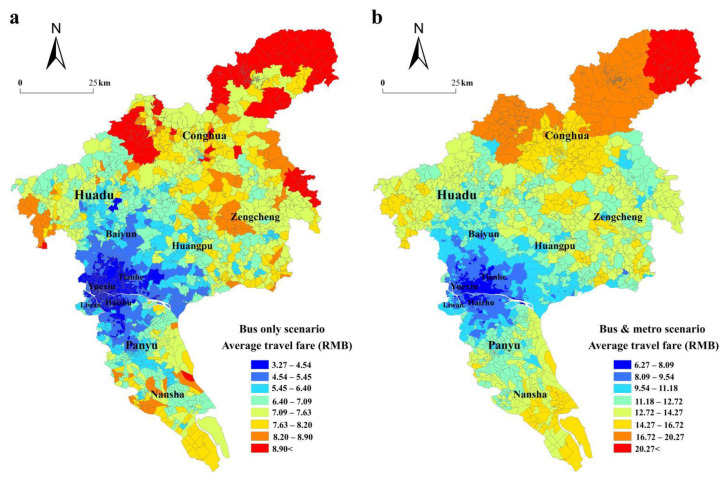
The spatial distribution of the transit fare of communities in the bus-only scenario (**a**), in the bus and rail transit scenario (**b**), and the fare changes between the two scenarios (**c**).

**Figure 6 ijerph-19-11428-f006:**
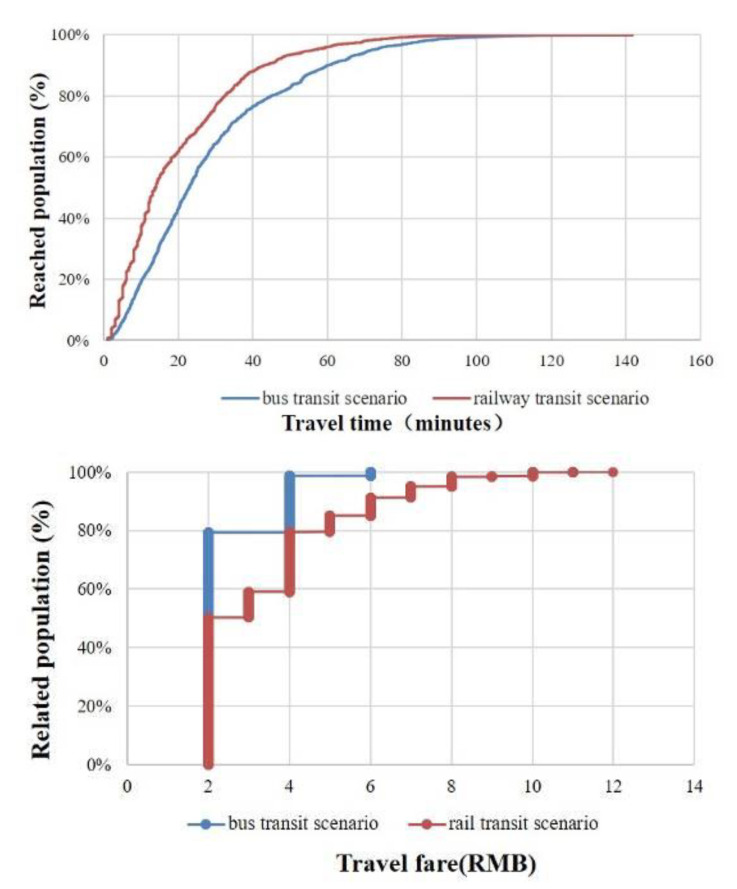
Cumulative share of the population accessing the closest urban commercial complex within a certain travel time and fare cost.

**Figure 7 ijerph-19-11428-f007:**
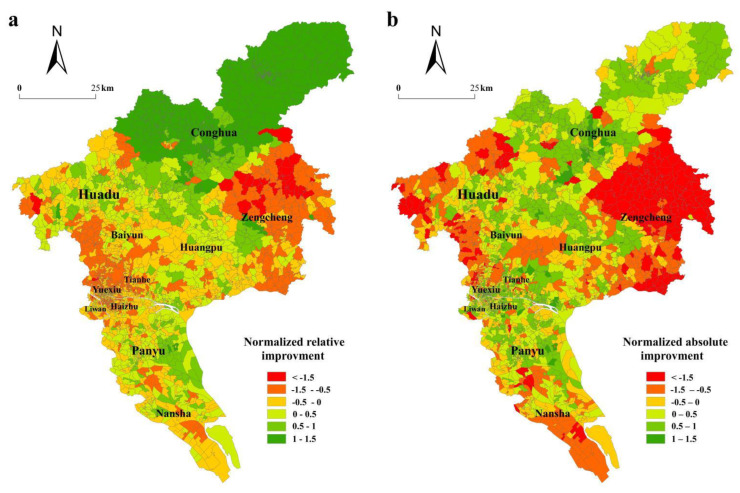
Normalized values of relative (**a**) and absolute (**b**) improvements in time-based accessibility.

**Figure 8 ijerph-19-11428-f008:**
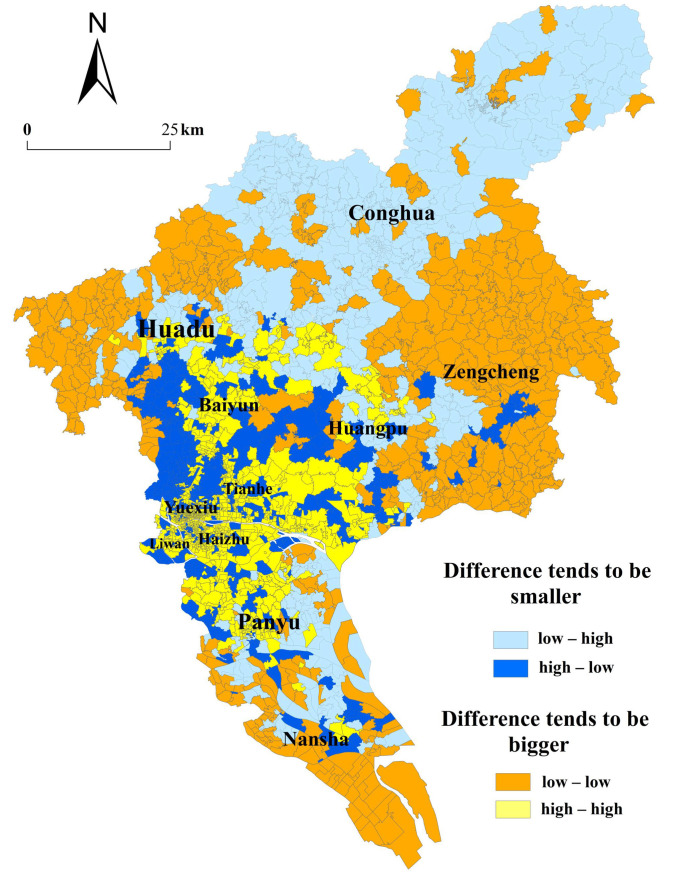
The distribution of the classification results of spatial equity by communities.

**Table 1 ijerph-19-11428-t001:** A comparison of the average travel speed and average transit fare in the bus-only and rail transit and bus scenarios.

Travel Mode	Average Travel Speed (km/h)	Average Transit Fare (RMB)
Bus only	18.25	5.51
Integrated bus transit and rail transit	27.95	12.12

**Table 2 ijerph-19-11428-t002:** The difference in the isochronous areal analysis of the CBD between the bus-only and bus and rail transit scenarios.

	Time Periods in the Bus-Only Scenario (Min)			Time Periods in the Bus and Rail Transit Scenario (Min)		
	0–30	30–60	60–120	0–30	30–60	60–120
Reachable population	78.94	307.27	510.94	258.93	691.49	2641.57
Spatial coverage area (km^2^)	41.32	224.32	1292.26	99.55	481.93	430.58

**Table 3 ijerph-19-11428-t003:** Coefficient of variation (*CV*) of time-based accessibility and fare-based accessibility.

		Number of Communities	Bus-Only Scenario			Bus and Rail Transit Scenario			Rate of Change	
			Standard Deviation	Mean Value	Coefficient of Variation	Standard Deviation	Mean Value	Coefficient of Variation	Mean Value	Coefficient of Variation
Time-based accessibility	All communities	2658	74.26	172.26	0.43	53.61	107.35	0.49	38.28%	0.28
	Inner Guangzhou	902	13.48	116.73	0.12	10.57	67.35	0.16	42.16%	0.17
	Middle Guangzhou	802	28.88	156.01	0.19	23.24	95.57	0.24	38.61%	0.25
	Outer Guangzhou	954	83.51	238.44	0.35	59.49	155.08	0.38	34.33%	0.38
Fare-based accessibility	All communities	2658	1.61	5.87	0.27	3.19	11.27	0.28	93.05%	0.23
	Inner Guangzhou	902	0.49	4.33	0.11	0.83	8.24	0.10	90.92%	0.18
	Middle Guangzhou	802	1.21	5.71	0.21	1.77	10.85	0.16	93.08%	0.26
	Outer Guangzhou	954	1.01	7.46	0.14	2.44	14.48	0.16	95.03%	0.26

**Table 4 ijerph-19-11428-t004:** The absolute improvements in time-based accessibility by population according to districts after rail transit lines were established.

z-Value Scores		<−1.5	−1.5–−0.5	−0.5–0	0–0.5	0.5–1.5	1.5<
Inner Guangzhou	Yuexiu district	0.00%	2.54%	3.67%	0.23%	0.00%	0.00%
	Haizhu district	0.00%	1.68%	6.32%	0.75%	0.09%	0.36%
	Liwan district	0.03%	1.42%	3.38%	0.57%	0.00%	0.00%
	Tianhe district	0.03%	2.70%	5.35%	0.86%	0.48%	0.54%
Middle Guangzhou	Baiyun district	0.19%	5.57%	6.15%	2.32%	0.25%	0.00%
	Huangpu district	0.00%	0.55%	2.40%	2.65%	0.77%	0.00%
	Panyu district	0.00%	0.58%	4.12%	5.30%	2.14%	0.08%
Outer Guangzhou	Huadu district	0.02%	0.25%	2.02%	2.72%	1.21%	0.02%
	Conghua district	0.00%	0.02%	0.01%	0.09%	1.20%	1.24%
	Zengcheng district	0.35%	2.50%	3.51%	1.27%	0.71%	0.02%
	Nansha district	0.00%	0.07%	1.15%	0.53%	0.29%	0.00%

**Table 5 ijerph-19-11428-t005:** The relative improvements in time-based accessibility by population according to districts after rail transit lines were established.

z-Value Scores		<−1.5	−1.5–−0.5	−0.5–0	0–0.5	0.5–1.5	1.5<
Inner Guangzhou	Yuexiu district	0.09%	0.97%	1.48%	2.36%	4.03%	0.01%
	Haizhu district	0.03%	0.47%	1.77%	5.02%	4.51%	0.36%
	Liwan district	0.18%	0.70%	1.56%	2.42%	2.13%	0.04%
	Tianhe district	0.18%	1.47%	2.17%	3.19%	4.20%	0.54%
Middle Guangzhou	Baiyun district	1.44%	5.40%	3.31%	3.83%	3.64%	0.20%
	Huangpu district	0.05%	0.86%	0.78%	2.35%	2.36%	0.30%
	Panyu district	0.32%	1.20%	2.99%	3.92%	4.03%	0.31%
Outer Guangzhou	Huadu district	0.37%	1.22%	1.32%	2.01%	1.60%	0.00%
	Conghua district	0.06%	0.14%	0.36%	0.81%	2.58%	0.87%
	Zengcheng district	2.45%	3.14%	1.78%	1.21%	0.77%	0.02%
	Nansha district	0.06%	0.70%	0.81%	0.27%	0.29%	0.00%

## Data Availability

Data are available upon request.

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
