# Peer review of "The Impact of Rail Transit on Accessibility and Spatial Equity of Public Transit: A Case Study of Guangzhou, China"

_ijerph, 2022, doi:10.3390/ijerph191811428_

Round 1

Reviewer 1 Report

The issues of transport accessibility and fairness in providing access to public transport are becoming increasingly relevant as urbanization grows and the negative impact of transport on the environment, which determines the relevance of the present study. The title of the article and keywords adequately reflect its content. The abstract is sufficient to understand the essence of the article, the state of the problem, gives an idea of the research methods, the results obtained.

In the introduction, the authors describe the features of bus route and rail transport, their impact on transport accessibility and fairness. The authors give questions, the solution of which will be devoted to the study and give the structure of the article. The second section includes a brief literature review of research on the topic of the article, confirming its relevance, including unsolved problems. In the third section, the authors describe the research methodology: characteristics of the research area, calculation of data and availability, method of equity analysis. The fourth section is devoted to a description of the results and analysis, including the impact of rail transport on accessibility, as well as the impact of rail transport on spatial equity. In the "Conclusions" section, the authors summarize their findings and results, pointing out how they can be used, and indicate directions for future research.

The article is prepared in accordance with the instructions for the authors, corresponds to the topic that it explores and publishes. Theoretical and practical conclusions are supported by figures and tables. The list of literary sources is sufficient in terms of the references number.

In our opinion, the article corresponds to the topic "improving transport accessibility” and corresponds in type to the preliminary study.

Article comments:

1. In our opinion, it is necessary to more clearly formulate the purpose and objectives of the study, as well as to confirm how the methods used make it possible to assess the adequacy of the conclusions given in the article, because, the concept of "transport accessibility" is closely related to the demand for travel, and spatial equity is usually regulated through tariff policy, which, in turn, is also related to passenger traffic.

2. It is necessary to clearly articulate how this study can help politicians, investors, the state and other stakeholders in the implementation of projects to expand the public transport network.

3. The list of references, in our opinion, can be expanded, since it poorly reflects the studies carried out in recent years and reflecting the use of new methods, models and tools for information mining to solve the problem formulated as a research topic, while this an area of research relevant to developing countries and economies.

4. The quality of some figures needs to be improved, because the inscriptions on them are hard to read.

Reviewer 2 Report

The authors compare minimal transit travel time and travel cost for the fastest paths between the center of each of 2000+ communities and each of eleven major commercial complexes within the Guangzhou metropolitan area. Two possible modes of travel are compared in regards to the travel time and cost-routes that, when possible, use the metro network and bus-based only routes. According to the flow chart in Figure 2, the calculations of the travel time somehow account for the transfers and for the waiting time at the initial and transfer stops, but this is not stated explicitly in the text.

Regrettably,  I see several serious flaws in the manuscript:

1. I see the basic methodological flaw in estimating accessibility-calculation of the average travel time to 11 commercial centers spread over the 100x100 km metropolitan area can be accepted as an exercise, but not as a result that reflects a human view of accessibility. Residents of the metropolitan North-West may be happy with the Conghua center and, generally, the distance to the nearest commercial center may reflect real residents’ behavior much better than the average. Saying nothing about average travel times of 7+ hours on the maps

2. I see the basic methodological flaw in interpreting differences in a trip cost-the trip that includes the metro is, on average, twice more expensive than the bus-only trip. But arent both trips cheap, at least when the nearest center is considered? Does this cost matter for the travelers at all? What is the average income of the travelers? How does it vary by the local communities? Are 2011 data good enough for estimating the cost of a trip and is it reasonable to merge them with the nowadays Baidu data?

3. Overall, the paper demonstrates that people residing at the periphery of the metropolitan area and aim at covering the entire metropolitan area with public transport, have to travel much longer and pay more than those residing in the center of the area. What else could be supposed?

4. Abstract-should be totally re-written. Currently, it starts with the trivial general statements and then dives into details that demand external knowledge
